# Genome-Wide Identification, Characterization, and Expression Profiling of *AP2/ERF* Superfamily Genes under Different Development and Abiotic Stress Conditions in Pecan (*Carya illinoinensis*)

**DOI:** 10.3390/ijms23062920

**Published:** 2022-03-08

**Authors:** Bingbing Yang, Xiaohua Yao, Yanru Zeng, Chengcai Zhang

**Affiliations:** 1The Research Institute of Subtropical Forestry, Chinese Academy of Forestry, Hangzhou 311400, China; BingbingYang1224@163.com (B.Y.); c.c.zhang@caf.ac.cn (C.Z.); 2State Key Laboratory of Subtropical Silviculture, College of Forestry and Biotechnology, Zhejiang A&F University, Hangzhou 311300, China

**Keywords:** ethylene-responsive elements (ERF), pecan (*Carya illinoinensis*), RNA-Seq, ERF-VII

## Abstract

The ethylene-responsive element (*AP2/ERF*) is one of the keys and conserved transcription factors (TFs) in plants that play a vital role in regulating plant growth, development, and stress response. A total of 202 *AP2/ERF* genes were identified from the pecan genome and renamed according to the chromosomal distribution of the *CiAP2/ERF* genes. They were divided into four subfamilies according to the domain and phylogenetic analysis, including 26 *AP2*, 168 *ERF*, six *RAV*, and two *Soloist* gene family members. These genes were distributed randomly across the 16 chromosomes, and we found 19 tandem and 146 segmental duplications which arose from ancient duplication events. The gene structure and conserved motif analysis demonstrated the conserved nature of intron/exon organization and motifs among the *AP2/ERF* genes. Several cis-regulatory elements, which were related to light responsiveness, stress, and defense responses, were identified in the promoter regions of *AP2/ERF*s. The expression profiling of 202 *CiAP2/ERF* genes was assessed by using RNA-Seq data and qRT-PCR during development (pistillate flowering development, graft union development, and kernel development) and under abiotic stresses (waterlogging, drought). Moreover, the results suggested that the ERF-VII members may play a critical role in waterlogging stress. These findings provided new insights into *AP2/ERF* gene evolution and divergence in pecan and can be considered a valuable resource for further functional validation, as well as for utilization in a stress-resistance-variety development program.

## 1. Introduction

Transcription factors (TFs), one of the master regulatory proteins types, perform a key role in the plant response to a range of abiotic and biotic stresses [1]. The APETALA2/Ethylene Responsive Factor (*AP2/ERF*) superfamily is one of the largest groups which regulates plant physiological processes, such as growth, development, and stress response [2,3]. The *AP2/ERF* transcription factors contain one or two highly conserved AP2 DNA binding domains with 60–70 amino acid residues that consist of three-stranded anti-parallel β-sheets, followed by a parallel α-helix [4]. The AP2 domain contains two conserved elements (YRG and RAYD). The YRG element is located at the N-terminal of the AP2 domain consisting of about 19–22 amino acid residues, and the RAYD element is located at the C-terminal consisting of about 43 amino acid residues [5]. The *AP2/ERF* TFs have been classified into the *AP2*, *ERF*, *RAV*, and *Soloist* subfamilies [6]. The *AP2* subfamily contains either single or double AP2 domains and plays an important role in the plant developmental processes [7]. The *ERF* subfamily was subdivided into the dehydration-responsive element-binding protein (*DREB*), and *ERF* subfamilies based on the amino acid sequences in the AP2 domain. The *ERF* subfamilies were further categorized into 12 groups: I–X, VI-L, and Xb-L [8]. The *RAV* subfamily consists of AP2 and B3 domains [9]. Additionally, a small group of TFs containing one AP2 domain, but a different structure, was classified as being in the *Soloist* subfamily [6].

With more extensive genome sequences, the AP2/ERF superfamily has been identified in various plant species, such as *Arabidopsis* [6,8], rice [8], poplar [10], oil palm [11], dark jute [12], cucumber [13], wheat [14], and sunflower [15]. These studies have shown that *AP2/ERF* superfamily members are involved in abiotic and biotic stress responses in plants. The AP2 subfamily binds to the GCAC(A/G)N(A/T)TCCC(A/G)ANG(C/T) element [16] and has been shown to be involved in organ development, including spikelet meristem differentiation, leaf epidermal cell designation, floral organ patterning, and seed yield [17,18,19]. Members of the RAV subfamily also have important functions in plant hormone signal transduction and the regulation of responses to biotic and abiotic stress by interacting with CAACA and CACCCTG in the promoter region [20,21]. The ERF subfamily members are generally involved in hypoxic stress response, pathogen stress response, injury response, and ethylene signaling pathway by binding a specific cis-acting component GCC-Box (sequence AGCCGCC) [22]. Moreover, DREB subfamily members generally bind to CRT/DRE elements (A/GCCGAC) to regulate stress response, including drought, salinity, and heat [23,24].

The pecan (*Carya illinoinensis* (Wangenh.) K. Koch) belongs to the Juglandaceae family and is an extremely important nut crop for daily diet in the world [25]. With the continuous expansion of planting area, the pecan is facing more and more adverse conditions, such as waterlogging, drought, and saline–alkali stress, so it is urgent to cultivate the resistant varieties. In the past three decades, traditional genetic methods, such as hybrid breeding, have contributed considerably to the production of new pecan varieties that increased productivity and stress resistance. However, the classical breeding technology is limited due to cross-incompatibility and its being time-consuming [26]. Current genetic-engineering methods, such as CRISPR/Cas9 [27] and TAILEN [28], have great potential for molecular improvement in short periods and high efficiency, but that requires the knowledge of the specific role of gene families.

To the best of our knowledge, the mechanisms underlying the regulation of pecan growth, development, and response to different environmental stresses remain poorly understood. Numerous reports have documented that the AP2/ERF-type TFs were important regulators involved in plant growth and development, as well as in hormonal regulation. For example, ERF-VII members play important roles in waterlogging tolerance in case of *Arabidopsis* and rice [29]. Overexpression of *RAP2.2*, *RAP2.12*, *HRE1*, and *HRE2* increased hypoxia tolerance in *Arabidopsis*, and *SUB1A-1* improved waterlogging tolerance in rice [30,31].

In the present study, the *AP2/ERF* TFs of pecan were investigated. The *AP2/ERF* genes structure, chromosome localization, conserved motif, phylogenetic analysis, duplication events, and cis-acting elements analysis were studied. Meanwhile, transcriptome data were used to analyze the expression of family members in various degrees of development and stress. Finally, five ERF-VII genes were selected to explore expression patterns under waterlogging stress, using real-time-PCR analysis. This study hopefully provides an important resource for further functional characterization of *CiAP2/ERF* genes and their utilization for the genetic improvement of the pecan.

## 2. Results

### 2.1. Identification, Classification, and Physical Properties of CiAP2/ERF TFs

A total of 202 *AP2/ERF* genes from the pecan genome were identified (Table 1 and Appendix A), and identified *AP2/ERF* genes were categorized into four families (*ERF, AP2, RAV,* and *Soloist*) depending on the occurrence of conserved AP2 domains.

Following the classification method in the *AP2/ERF* factors of Arabidopsis [8], these genes were further analyzed and classified into two subfamilies (*ERF* and *DREB*) according to their specific amino acid variation in the conserved domain region. The *ERF* subfamily contained 107 members with alanine (A) in the 14th and aspartic acid (D) in the 19th positions, while the *DREB* subfamily had 61 members having valine (V) and glutamine (E) residues at respective positions. Twenty-six genes were classified into the *AP2* family, including 23 genes containing two AP2 domains and 3 genes with only one AP2 domain. The *RAV* family had 6 genes with the AP2 and B3 domain. Moreover, 2 specific genes contained a single domain and showed the highest homology with the At4g13040 gen; they were in the *Soloist* family (Table 1 and Appendix A).

The length of *CiAP2/ERF* genes ranged from 387 to 2283 bp, and the proteins contained 128–760 of amino acids length. The molecular weight (MW) and Isoelectric point (PI) varied widely, which was 13906.60–82684.48 and 4.47–10.73, respectively. Predicted subcellular localization revealed that a majority of *AP2/ERF* genes were located in the nucleus and cytoplasm (Appendix A).

### 2.2. Phylogenetic, Gene Structure, and Conserved Motif Analysis of CiAP2/ERF TFs

A phylogenetic tree was constructed by using the deduced protein sequences of *AP2/ERF* genes from pecan and Arabidopsis that were clustered into distinct ERF, AP2, RAV, and Soloist clades (Figure 1). Most of the members were classified into the *ERF* family. In-depth analysis subdivided the ERF clades into twelve sub-clades (I–X, Xb-L, and VI-L), similar to gene classification in Arabidopsis. The first four sub-clades (I–IV) were grouped as *DREB* subfamily, and the remaining eight sub-clades (V–X, Xb-L, and VI-L) were in the *ERF* subfamily. Interestingly, *CiAP2-26* was assigned to group ERF-IV (Figure 1).

The number of introns varied in different subfamilies of the *AP2/ERF* superfamily. Except *CiAP2-26*, which had 1 intron, the AP2 family members contained more introns (4–10). Most of *ERF* family members did not have any introns, and a few of them had 1 or 2 introns. *CiRAV-05* had 3 introns, and the other *RAV* family members had no introns. The remaining two *Soloist* family members (*CiSoloist1* and *CiSoloist2*) contained 6 and 7 introns, respectively (Figure 2D). In general, closely related members from the same subfamily had similar numbers and intron lengths. Furthermore, the conserved motifs of the *AP2/ERF* family members were studied. In total, 10 conserved motifs were presented and designated Motifs 1 to 10 (Figure 2B and Appendix A). Motifs 1 and 2 were presented in almost all sequences except *CiAP2-26*, which contained Motifs 3, 5, 7, and 9. Most *ERF* family members had Motifs 1, 2, and 3; most *DREB* family members contained Motifs 1, 2, 3, 4, and 10; the majority of *AP2* family members and all *RAV* family members had Motifs 1, 2, 3, 4, and 5; all *Soloist* family members contained Motifs 1, 2, and 4 (Figure 2B).

Based on the presence, position, and phase of introns, the intron distribution within the AP2 domain of all of the reported *AP2/ERF* genes was analyzed. Eleven different distribution patterns (designated A to K), ranging from zero to seven introns within the domain, were observed. (Figure 3). The results showed that 81.7% identified members of the family did not have introns in their AP2 domain, thus indicating that the members were conserved. All *DREB* and *RAV*, and most *ERF* subfamily members belonged to Pattern A, and the remaining ERF members belonged to Pattern B. *Soloist* subfamily members had Pattern C. Except for *CiAP2-26*, other *AP2* subfamily members covered from Pattern D to Pattern K (Figure 3 and Appendix A).

### 2.3. Chromosomal Distribution and Genes Duplication Analysis of 202 CiAP2/ERF TFs

All 202 identified *CiAP2/ERF* genes were mapped to 16 chromosomes of pecan, and the distribution of *CiAP2/ERF*s varied through 16 chromosomes (Figure 4). Chromosome 1 anchored the highest number of 22 *AP2/ERF* genes, followed by Chromosome 3 (20) and Chromosome 4 (20). Chromosome 16 had the lowest number (6) of *CiAP2/ERF*s. *ERF* and *DREB* subfamily members were distributed on all 16 chromosomes of pecan. However, none of the *AP2* subfamily members were mapped to Chromosome 3, 4, 5, 11, or 12. Moreover, *RAV* subfamily members were only mapped to Chromosomes 4, 13, 14, 15, and 16. At the same time, *Soloist* subfamily members were located only on Chromosomes 1 and 2 (Figure 4).

The calculated Ka/Ks ration values of all *AP2/ERF* genes were found to be less than 1 (Appendix A). These results indicated that the evolution of pecan *AP2/ERF* genes happened via strong purifying selection. Moreover, a total of 165 pairs of gene duplications were identified and divided into tandem duplications and segmental duplications. Among them, 19 pairs were identified as tandem duplication, and 146 pairs were classified as segmental duplication (Figure 4 and Figure 5). All 16 chromosomes of pecan genome contained segmental duplicated *AP2/ERF* family members. Ten chromosomes (1, 3, 4, 7, 8, 9, 10, 11, 12, and 13) had tandem duplication, whereas Chromosome 8 contained more (Figure 4 and Figure 5). The tandem and segmental duplication is the driving force for gene-family expansion in plants [32]. These results demonstrated the prominent role of tandem and segmental duplication in the expansion of *AP2/ERF* gene family members in the pecan genome.

### 2.4. Analysis of Cis-Acting Elements in Promoter Regions of CiAP2/ERFs

To investigate the cis-elements of the 202 *CiAP2/ERF* genes, 2000 bp of sequence upstream from the start codon was analyzed. The cis-elements of the *CiAP2/ERF* genes contained 26 categories, which were related to plant growth and development, stress response, and plant hormone response (Figure 6 and Appendix A). Light response elements were found in the largest number in all *AP2/ERF* genes’ promoter sequences, followed by abscisic acid responsive elements and MeJA-responsive elements, which were also identified in most members (Figure 6 and Appendix A).

### 2.5. RNA-Seq Expression Analysis of CiAP2/ERF Genes in the Response of Stress and Development

RNA-Seq analysis was used to identify the function of *AP2/ERF* genes in different development and abiotic stress conditions. For the drought condition, 0 (control), 3, 6, 9, 12, and 15 days were considered where 19 genes were not expressed (Figure 7A). Among the 26 *AP2* subfamily members, 10 had the highest expression at 3 days, and 8 showed the highest expression at 15 days. This indicated that these genes might be induced at 3 and 15 days. Most *ERF* members were upregulated at 15 days, while *DREB* members showed a similar trend. Three *RAV* members had the h ighest expression at 6 days, while *CiRAV-05* had the highest expression at 0 days. The *Soloist* subfamily was mainly expressed at 12 and 15 days (Figure 7A).

In the pistillate flowering development, FB1 indicated the initial stage of flower-bud differentiation, FB2 indicated the inflorescence formation stage, FB3 indicated that the female flower primordium continues, FL1 indicated the early blooming stage, and FL2 indicated the full blooming stage (Figure 7B). There were 20 genes that were not expressed. Most *AP2* subfamily genes were highly expressed at the FL stage, while the majority of *ERF* and *DREB* genes was upregulated at the FB stage. Particularly, most members were highly expressed in the FB3 stage. Except for *CiRAV-02* and *CiRAV-03*, the other members of *RAV* subfamily were also highly expressed at the FB3 stage.

In the graft union development, nine genes were not expressed. Most *AP2* subfamily genes were mainly upregulated at 0 and 8 days, while the other subfamily members were mainly upregulated at 15 and 30 days (Figure 7C). Similarly, in the kernel development (Figure 7D), thirty-five genes were not expressed. *AP2* subfamily genes were mainly upregulated at the later water stage and gel stage. Most *ERF* genes were upregulated at the later water stage, while *DREB* genes were upregulated at the later water and mature stage. *RAV* and *Soloist* subfamily members showed similar expression patterns which were upregulated at the late water stage and mature stage.

### 2.6. ERF-VII Members in Pecan and Arabidopsis

There were five group ERF-VII genes in the Arabidopsis genome, namely *AtRAP2.2*, *AtRAP2.3*, *AtRAP2.12*, *AtHRE1*, and *AtHRE2*. Similarly, five members of ERF-VII were identified in the pecan genome: *CiERF-005*, *CiERF-010*, *CiERF-020*, *CiERF-052*, and *CiERF-064*. The amino acid sequences of these ten genes were used for multi-sequence alignment and tree construction. The resultant showed that these ERFs are characterized by a conserved DNA-binding domain by the extremely conserved N-terminal motif MCGGAI/VI (Figure 8). In order to estimate the evolutionary relatedness among the members of this group, a neighbor-joining method was used to build a phylogenetic tree of the ERF-VII proteins from Arabidopsis and pecan. *CiERF-010*, *CiERF-020*, *AtRAP2.12*, and *AtRAP2.2* cluster together. In the same way, *CiERF-005* and *AtRAP2.3* cluster together. *CiERF-052*, *CiERF-064*, and *AtHRE2* cluster together.

### 2.7. ERF-VII Genes’ Expression Analysis of Different Pecan Cultivars under Waterlogging Stress

To further explore the response of *CiAP2/ERF* genes under waterlogging stress, the expression patterns of 5 *CiAP2/ERF* genes (*CiERF-064, CiERF-052, CiERF-020, CiERF-010,* and *CiERF-005*) from ERF-VII group were examined via RT-qPCR analysis (Figure 9). In the case of waterlogging stress, *CiERF-064* was significantly upregulated before 12 h in all varieties except C10, which was downregulated at 4 h. *CiERF-052* followed downregulation trends but moved up at 12 h. In addition to J23, *CiERF-020* showed a trend of first rising and then decreasing. In comparison to other varieties, it showed a trend of first decreasing and then increasing. *CiERF-010* was downregulated at 24 h. *CiERF-005* was downregulated before 24 h and upregulated at 48 h. (Figure 9). Thus, these experiments indicated that ERF-VII genes might have a strong correlation with waterlogging stresses.

## 3. Discussion

The *AP2/ERF* is one of the major transcription factor families in plants. This gene family plays a key role in complex growth processes, including seed germination, flower growth, leaf senescence, response to biotic, and abiotic stress [6,33,34,35]. Genome-wide analysis based on genome sequence can provide a basis for the comprehensive identification of the *AP2/ERF* family and an understanding of the function and evolution of *AP2/ERF* factors [36]. To date, the *AP2* subfamily of pecan has been studied [37]. However, genome-wide studies have not been conducted to explore the structure and role of *AP2/ERF* superfamily genes in the pecan. Therefore, we report the *AP2/ERF* transcription factors in pecans from the whole genome. In addition, we also explored their role in different pecan cultivars under waterlogging stress based on real-time PCR analysis and the function of development and stress (drought, pistillate flowering development, graft union development, and kernel development), using transcriptome data.

In our study, 202 *AP2/ERF* family genes of pecan were identified. Compared with other plants, the total number of *AP2/ERF* genes in the pecan is greater than that in many plants, but less than the sunflower (288) [15], *Brassica rapa* (281) [38], and sugarcane (218) [39]. Previous studies reported that the *AP2/ERF* number was increased with the plant evolution or whole-genome duplication [40], which may be in close relationship with functional specificity in plants. The numbers of *AP2/ERF* genes varied significantly in these plants that suggested their expansion during evolution. The wide variation of molecular weight (13906.60–82684.48) and isoelectric point (4.47–10.73) indicated the putative novel variants in *CiAP2/ERF* genes. Predicted subcellular localization suggested that all *AP2/ERF* members were located in the nucleus and cytoplasm. This suggested that these genes might function in transcription and post-transcriptional modification.

The number and position of introns give clues on evolutionary relationships of proteins. The pecan *AP2/ERF* gene structural analysis revealed that a higher number of *AP2* subfamily members contained more introns (4–10) compared to *ERF, RAV*, or *Soloist* subfamily members. This was consistent with the results observed in other plants [11,15,41,42]. It suggested that the *AP2/ERF* superfamily members of pecan were functionally diverse, but the functions of members of the same subfamily were similar, which further proved the accuracy of *AP2/ERF* genes evolution and classification. Introns can be divided into three types based on phases: Phase 0, Phase 1, and Phase 2 [43]. An exon flanked by two introns of the same phase was called a symmetrical exon, and exons that were subjected to alternative splicing were always symmetrical exons [43]. Intron distribution of AP2 domain suggested that *ERF, RAV, DREB*, and *Soloist* subfamily members were more conserved than *AP2* subfamily members in pecan.

In addition, conserved motifs analysis was used to determine transcriptional activity, protein–protein interactions, and DNA-binding activity of TFs [44]. The conserved motifs analysis of pecan *AP2/ERF* TFs indicated that these motifs might be related to specific functions. Gene-duplication events are one of the main evolutionary mechanisms for the generation of new genes and an important way to cope with the environmental changes during growth and development stages [45]. Our study demonstrated the occurrence of tandem and segmental duplications in pecan for the expansion of all *AP2/ERF* superfamily members during evolution. The Ka and Ks measure the course of divergence, and the ratio of Ka/Ks was a checkpoint for positive selection pressure after duplication. The Ka/Ks > 1 means accelerated evolution with positive selection, Ka/Ks = 1 stands for neutral selection, and Ka/Ks < 1 represents purifying selection [46]. The mean value of Ka/Ks (0.239) of pecan suggested a purifying selection of the *AP2/ERF* genes that underwent great selective constraint and substitutions elimination by natural selection.

Gene functions can be preliminarily predicted by analysis of the gene-expression profiles [47]. *DREB2A*, *DREB2B* [48], and *CBF4* (*DREB1D*) [49] were induced by drought stress in Arabidopsis. Similarly, drought transcriptome data suggested that *AP2/ERF* genes were upregulated at 3 and 15 days in pecan. Three *AP2/ERF* family genes, *PhAp2A*, *PhAp2B*, and *PhAp2C*, were isolated from petunia and were verified to perform function in flower development [50]. In pecan, transcriptomic data of flower development also suggested that the *AP2/ERF* genes may play a key role in early and middle stages. Previous research showed that *ERF* subfamily members, specifically the ERF-VII members, were involved in the submergence stress response. For example, water-tolerance gene, *Sub1A*, restricts rice elongation at the seedling stage during flash floods [31]. In addition, there were five genes in the ERF-VII of Arabidopsis, which were *RAP2.2 (At3g14230), RAP2.12(At1g53910), RAP2.3 (At3g16770), HER1 (At1g72360),* and *HER2 (At2g47520)*. *RAP2.2*, *RAP2.12*, *HRE1*, and *HRE2* were proved to play vital roles in hypoxia response [30,51]. Similarly, five ERF-VII members in pecan were identified. In Arabidopsis, the stability of most ERF-VII proteins was regulated by the N-end rule pathway [52]. Conserved motif MCGGAI/VI in N-terminal of ERF-VII genes in pecan were also found. This suggested that members of ERF-VII in pecan and Arabidopsis may perform similar functions. Then the quantitative analysis for ERF-VII genes in pecan by qRT-PCR suggested that their expression levels also changed significantly with the time of flooding in pecan.

Overall, the abovementioned findings provided insights into the potential functional roles of pecan *AP2/ERF* genes. The comprehensive analyses were beneficial to screening candidate *AP2/ERF* genes for further functional characterization, as well as for the genetic improvement in the agronomic characters of pecan.

## 4. Materials and Methods

### 4.1. Identification of the CiAP2/ERF TFs in Pecan Genome

The whole-genome sequences and deduced protein sequences of pecan (*Carya illinoinensis*) were downloaded from the NCBI (https://www.ncbi.nlm.nih.gov/, accessed on 6 September 2021). The well-studied AP2/ERF protein sequences of *Arabidopsis thaliana* were downloaded from the PlantTFDB (http://planttfdb.gao-lab.org/, accessed on 6 September 2021) database and were employed as a query (e-value ≤ 1 × 10^−10^ and identity ≥ 50%) to search the homologous proteins in the pecan genome through BLASTP. The Hidden Markov Model (HMM) profiles of AP2 domains (PF00847) were obtained from the Pfam database (https://pfam.xfam.org/, accessed on 6 September 2021), and AP2/ERF genes were retrieved from the pecan genomic database by HMMER3.0. The candidate sequences were further verified, and redundant sequences were removed from the list. Furthermore, output putative AP2/ERF protein sequences were submitted to HMMER (https://www.ebi.ac.uk/Tools/hmmer/, accessed on 6 September 2021), CDD (https://www.ncbi.nlm.nih.gov/Structure/bwrpsb/bwrpsb.cgi, accessed on 6 September 2021), Pfam (https://pfam.xfam.org/, accessed on 6 September 2021), and SMART (http://smart.embl-heidelberg.de/, accessed on 6 September 2021) to confirm the conserved AP2 domain. Finally, a total of 202 CiAP2/ERF genes were identified from the genome. ExPASy (https://web.expasy.org/protparam/, accessed on 6 September 2021) was used to determine the isoelectric point (pI) and molecular weight (MW) of AP2/ERF proteins. Subcellular localization of *AP2/ERF* genes was predicted by using the Cell-PLoc2.0 (http://www.csbio.sjtu.edu.cn/bioinf/Cell-PLoc-2/, accessed on 6 September 2021).

### 4.2. CiAP2/ERF Gene Structure, Chromosome Localization, and Conserved Motif Analysis

The conserved motifs of *AP2/ERF* genes were searched by using MEME (https://meme-suite.org/meme/tools/meme, accessed on 6 September 2021) with default settings, and the gene structure map was obtained with TBtools software [53]. The combined visualization of the phylogenic tree, the conserved motifs, and the gene structures of the *CiAP2/ERF*s were also conducted with TBtools [53]. The chromosomal distribution of 202 *CiAP2/ERF*s was also investigated against the pecan genome, using the TBtools [53].

### 4.3. Phylogenetic Analysis and Duplication Events Analysis of CiAP2/ERF Genes

The evolutionary relationships among AP2/ERF protein sequences of pecan (202 *AP2/ERF*s) and *Arabidopsis*
*thaliana* were established by constructing a phylogenetic tree via the maximum likelihood method, using IQ-TREE version 1.6.12 software. Best-fit model according to BIC (VT + F + R10) was used to construct the phylogenetic tree. Finally, iTOL (https://itol.embl.de/itol.cgi, accessed on 6 September 2021) was used to visualize the evolutionary tree. *AP2/ERF* gene duplication events were analyzed by using the MCScanX tool (Multiple Collinearity Scan) with a set of parameters.

The evolutionary pattern and the divergence of homologous genes were estimated by computing the synonymous (Ks) and nonsynonymous (Ka) substitution. The TBtools [53] software was used to calculate Ka and Ks values of pecan.

### 4.4. Cis-Acting Elements Analysis of CiAP2/ERFs

The cis-acting regulatory elements in the 2000 bp genomic sequence upstream of the coding *CiAP2/ERF* gene sequences were investigated by using the online PlantCARE databases (http://bioinformatics.psb.ugent.be/webtools/plantcare/html/, accessed on 6 September 2021).

### 4.5. Transcriptome Data Sources and Expression Analysis of the AP2/ERF Genes

The transcriptome data of pecan (drought, pistillate flowering development, graft union development, and kernel development) were downloaded from the SRA (Sequence Read Archive) database of the NCBI website. Kernel development data were downloaded from accession number PRJNA504494 that were previously generated by our group [25]. Transcriptome data of graft union development were retrieved from SRA accession number PRJNA411951 [54]. Pistillate flowering development transcriptome data were considered, and the raw data were retrieved under the accession number PRJNA533506 [55]. Moreover, drought transcriptome data were downloaded from the SRA database under accession number PRJNA743302 [56]. All the transcriptome sequencing data obtained from Illumina sequencing were quality checked by FastQC v.0.11.9 (http://www.bioinformatics.babraham.ac.uk/projects/fastqc, accessed on 6 September 2021). The raw data were filtered to trim low-quality reads by Trimmomatic v.0.36 [57]. These trimmed high-quality reads were used in further downstream analysis. The HISAT v.2.1.0 [58] was used to map the paired-end trimmed reads to the pecan genome database deposited in NCBI. The transcript abundance of *CiAP2/ERF*s in different transcriptome was calculated with FPKM (Fragments Per Kilobase of exon model per Million mapped fragments) values. The TBtools was used to generate a heatmap of all 202 *CiAP2/ERF*s.

### 4.6. Pecan Plant Materials and Stress Exposure

The pecan (*Carya illinoinensis*) plantlets were raised at the test site at the Research Institute of Subtropical Forestry, Hangzhou, China. Healthy pecan grafting plantlets of the same age (13 months old) were chosen to undergo waterlogging. Before stress exposure, all plants were shifted to a growth chamber at 27 °C for one week. Waterlogging treatment was conducted after 0 (control), 4, 12, 24, and 48 h, using different plantlets (3 for each point in time). At different time intervals (0, 4, 12, 24, and 48 h) of the flood-water-stress exposures, the mature leaves were collected and immediately frozen in liquid nitrogen for further qRT-PCR analysis.

### 4.7. Quantitative Real-Time PCR (qRT-PCR) Analysis of CiAP2/ERFs

Five genes were selected for expression analysis of four pecan varieties (Mahan, C10, C20, and J23). The primers of these genes were designed by PrimerExpress software v3.0.1 (Applied Biosystems, Foster City, CA, USA). Electrophoresis with 1% agarose and melting-curve analysis was used to calculate specificity during quantitative real-time PCR analysis.

For RNA extraction, samples (2 g) were crushed into a fine powder with liquid nitrogen and mortar–pestle. Then total RNA was isolated from the leaf samples of the above waterlogging treatment by using a Tiangen RNAprep plant kit (Tiangen, Beijing, China), and gel electrophoresis was used to check RNA integrity. First-strand complementary DNA (cDNA) was prepared with PrimeScript™ RT Master Mix (TaKaRa) as per the manufacturer’s instructions. A 20 μL final reaction was prepared in a 96-well plate containing 10 μL of TB Green Premix Ex Taq II (Tli RNaseH Plus) (2X), 0.4 μL of ROX Reference Dye, 2 μL cDNA, 6.8 µL ddH2O, and 0.4 μL of each primer. The quantitative RT-PCR expression was carried out with a QuantStudio 7 Flex Real-time PCR system (Applied Biosystems) with the following steps: (1) denaturation stage, 95 °C/30 s; and (2) PCR stage, 95 °C/3 s, 60 °C/30 s for 40 cycles. All the amplifications were performed with three repeats. The expression analysis was calculated by using the comparative CT method (2^−ΔΔCt^) [59]. The list of qPCR primers used in this study is provided in Appendix A. One-way analysis of variance (ANOVA) was used to determine the statistical significance at the *p* ≤ 0.05, *p* ≤ 0.01, *p* ≤ 0.001, and *p* ≤ 0.0001 levels.

## 5. Conclusions

A total of 202 *AP2/ERF* genes were identified and analyzed through conserved domain, motif distribution, chromosomal distribution, genes duplication, and cis-acting elements. These results help to classify this gene family and provide insight into *AP2/ERF* genes’ evolution. Furthermore, the role of *AP2/ERF* genes in various development and abiotic stress conditions, using transcriptome data and quantitative experiments, was analyzed. The results suggest that members of ERF-VII in pecan may perform similar functions as members of Arabidopsis. In short, these results provide a valuable resource for better understanding the biological roles of *AP2/ERF* genes in the pecan.

## Figures and Tables

**Figure 1 ijms-23-02920-f001:**
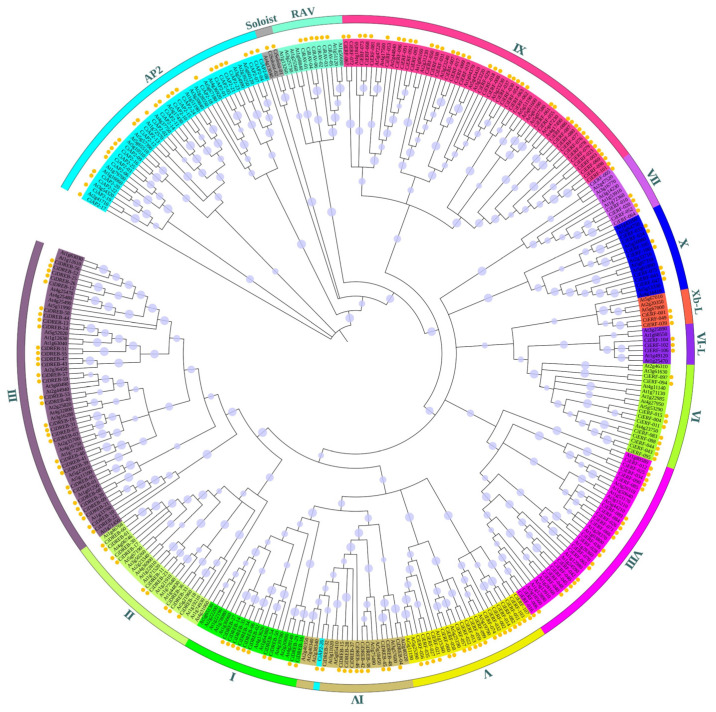
An unrooted phylogenetic tree constructed by the maximum likelihood method, using AP2/ERF proteins of pecan and Arabidopsis. All of the families from pecan and Arabidopsis are grouped and represented in various colors.

**Figure 2 ijms-23-02920-f002:**
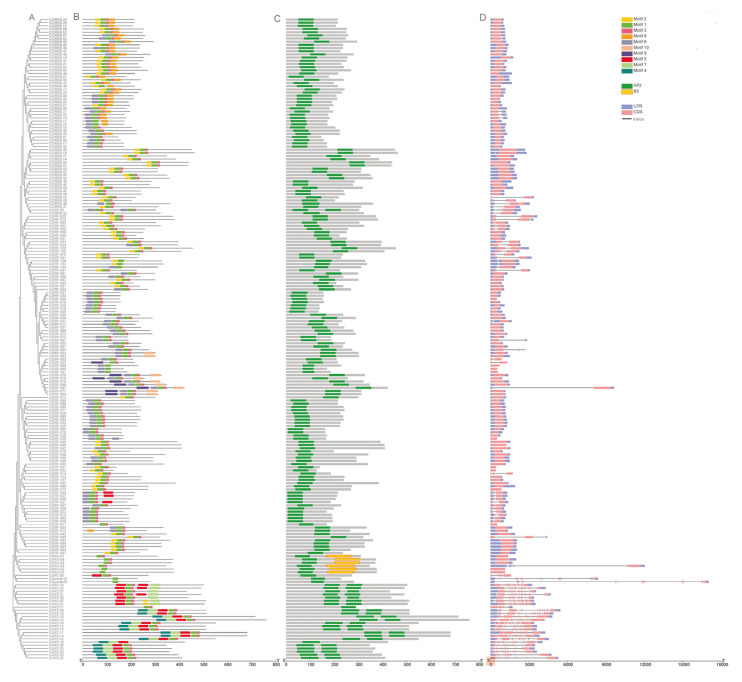
Gene structure, conserved domain, and motifs of *AP2/ERF* genes according to the phylogenetic relationship. (**A**) The phylogenetic tree was conducted based on the full-length sequences of pecan 202 AP2/ERF proteins, using IQ-TREE software. (**B**) The motif composition of pecan AP2/ERF proteins. The motifs, numbers 1–10, are displayed in different colored boxes. (**C**) The AP2 domains are highlighted by green boxes and B3 domain by yellow boxes. (**D**) Exon–intron structure of pecan AP2/ERF genes. Blue boxes indicate untranslated 5′- and 3′-regions; pink boxes indicate exons and black lines indicate introns.

**Figure 3 ijms-23-02920-f003:**
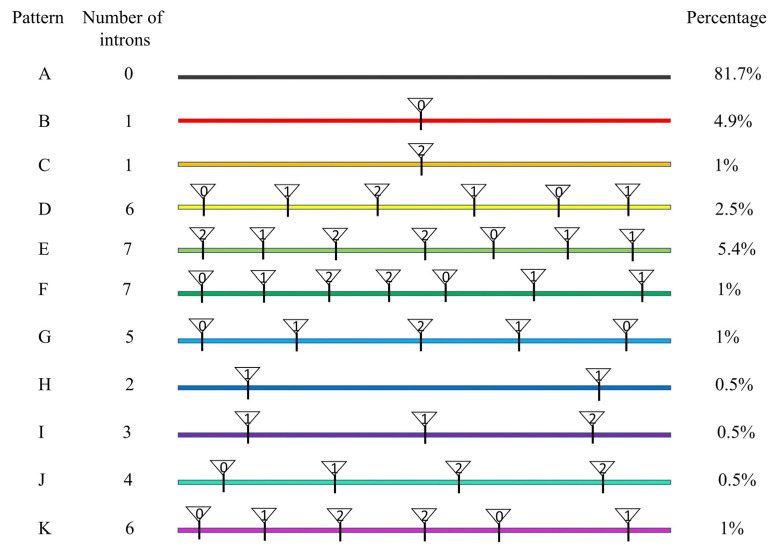
Intron distribution within the AP2 domains of the *CiAP2/ERF* genes. Scheme of the intron distribution patterns (color-coded and designated **A**–**K**) within the AP2 domains of the CiAP2/ERF proteins. White triangles and numbers within them indicate the locations of introns and splicing phases (0, Phase 0; 1, Phase 1; and 2, Phase 2), respectively. The percentage of CiAP2/ERF proteins in each pattern is shown on the right.

**Figure 4 ijms-23-02920-f004:**
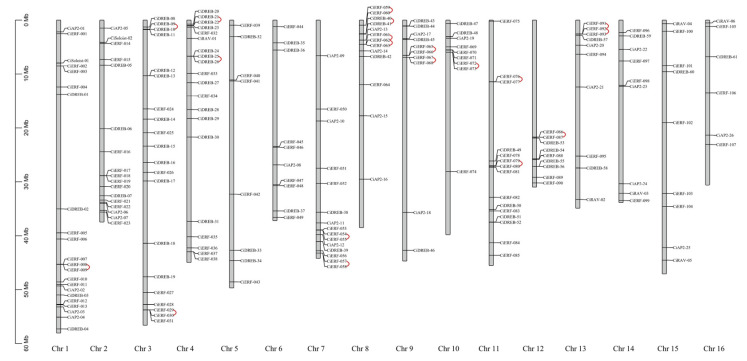
Chromosomal mapping of 202 pecan *AP2/ERF*s. The length of the pecan chromosomes is indicated on the vertical grayscale. The chromosome numbers (1–16) are indicated on the bottom of each chromosome. Red lines indicate tandem duplication.

**Figure 5 ijms-23-02920-f005:**
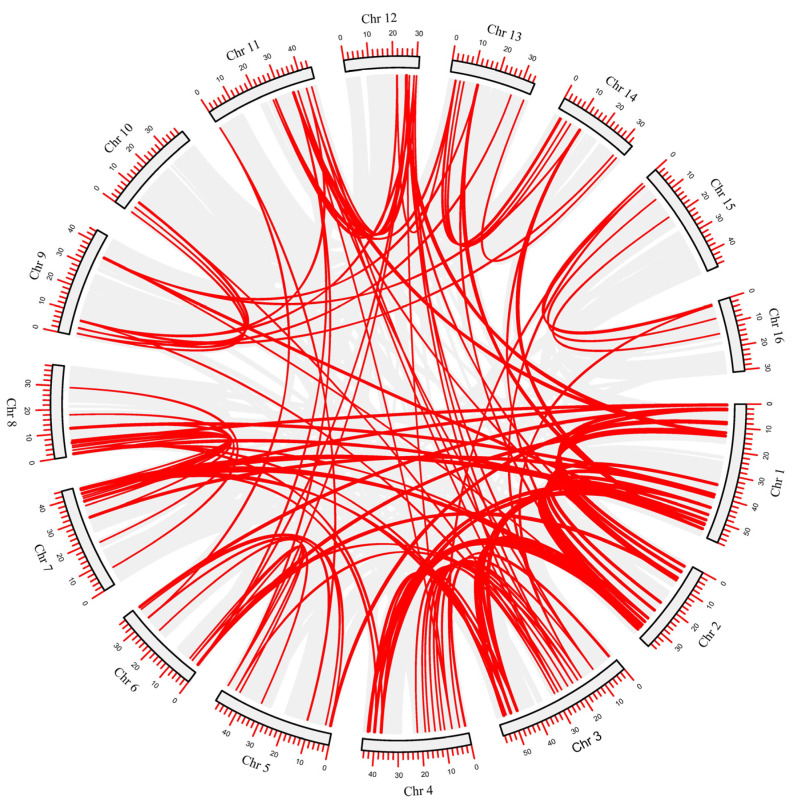
Schematic representations of the inter-chromosomal relationships of pecan *AP2/ERF* genes. The red lines indicate duplicated *AP2/ERF* gene pairs in pecan. The chromosome number is indicated in the middle of each chromosome.

**Figure 6 ijms-23-02920-f006:**
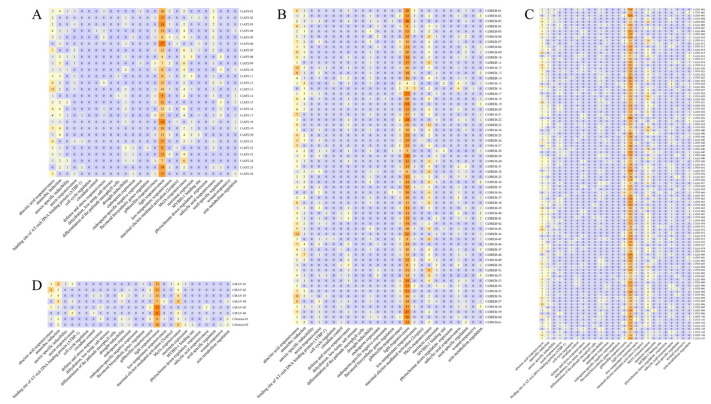
Cis-elements of *AP2/ERF* genes in pecan: (**A**) AP2 subfamily, (**B**) DREB subfamily, (**C**) ERF subfamily, and (**D**) RAV–Soloist subfamily. The numbers of different cis-elements in the *CiAP2/ERF* genes are indicated in different colors in the grid.

**Figure 7 ijms-23-02920-f007:**
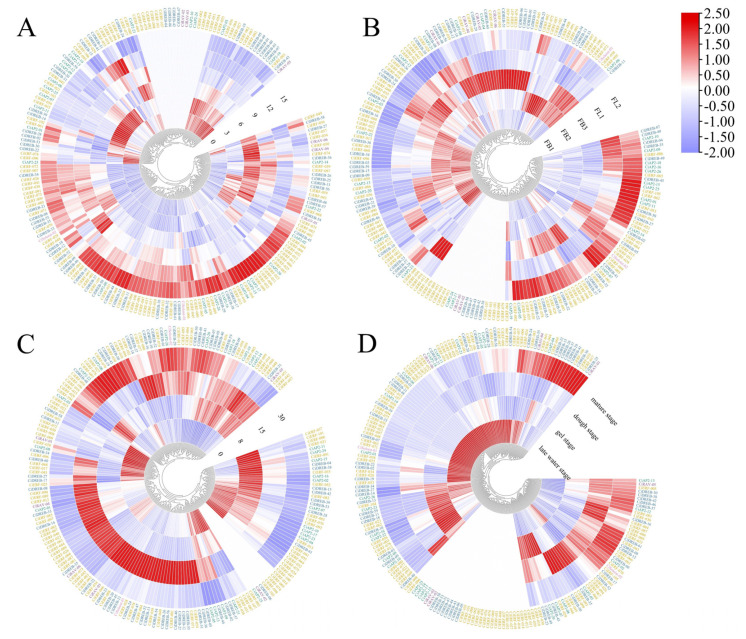
Expression profiling of *CiAP2/ERF* genes: (**A**) drought, (**B**) pistillate flowering development, (**C**) graft union development, and (**D**) kernel development. Gene expression was measured by quantified transcription levels (fragments per kilobase of exon model per million mapped reads, FPKM) derived from RNA-Seq analysis. Green indicates *AP2* subfamily, brown indicates *ERF* subfamily, blue indicates *DREB* subfamily, violet indicates *RAV* subfamily, and pink indicates *Soloist* subfamily.

**Figure 8 ijms-23-02920-f008:**
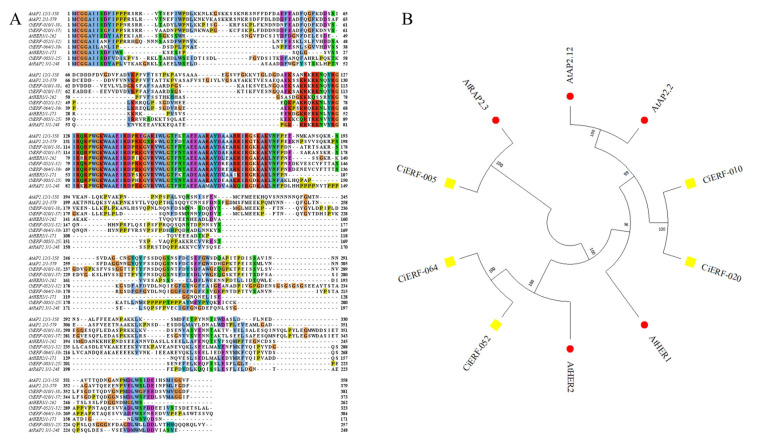
Group VII ethylene responsive factors (ERFs) in pecan and Arabidopsis. (**A**) Alignment of the protein sequences of the pecan and Arabidopsis group VII ERFs. Amino acidic sequences were aligned by using ClustalX2 software. (**B**) Phylogenetic tree illustrating the relatedness of group VII ERFs among pecan and Arabidopsis. An unrooted neighbor-joining tree was created by using MEGA11.

**Figure 9 ijms-23-02920-f009:**
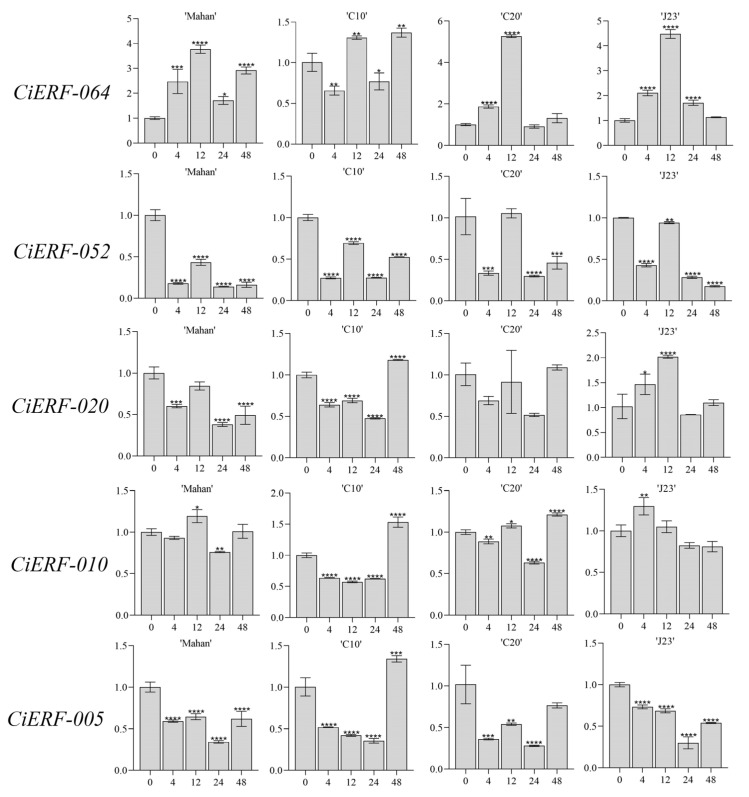
Real-time PCR analysis of 5 *AP2/ERF* genes in different pecan cultivars under waterlogging stress (0, 4, 12, 24, and 48 h) conditions. *X*-axis represents the hours of stress exposure. *Y*-axis represents the relative expression of a transcription factor. Data represent the mean ± SE of three replicates. Asterisks represent significant difference at *p* ≤ 0.05 (*), *p* ≤ 0.01 (**), *p* ≤ 0.001(***), and *p* ≤ 0.0001 (****).

**Table 1 ijms-23-02920-t001:** Summary of the AP2/ERF superfamily in pecan genome.

Classification	Group	No.
AP2 family	Double AP2/ERF domain	23
	Single AP2/ERF domain	3
DREB family	I	10
	II	9
	III	32
	IV	10
ERF family	V	19
	VI	10
	VI-L	3
	VII	5
	VIII	21
	IX	39
	X	7
	Xb-L	3
RAV family		6
Soloist family		2
	Total	202

## Data Availability

Not applicable.

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
