# Peer review of "Genome-Wide Identification, Characterization, and Expression Profiling of AP2/ERF Superfamily Genes under Different Development and Abiotic Stress Conditions in Pecan (Carya illinoinensis)"

_ijms, 2022, doi:10.3390/ijms23062920_

Round 1

Reviewer 1 Report

The publication proposal has been written in a clear and transparent manner that does not raise any doubts. The first part introduces the reader to the problem well. Methodically, the study was well prepared and quite well described. Unfortunately, the authors did not avoid mistakes, there is a lot of unnecessary signs “ \ “ in the places that should be “ / “ etc. I recommend that you carefully check the text and introduce editorial corrections.

Reviewer 2 Report

This is a well-written article with a fairly standard set of bioinformatics techniques for describing a family of genes and proteins in a single plant species. This plant was an agronomically important pecan tree.
Remarks are displayed as comments in the attached file.
I note the low resolution and quality of most of the drawings. And that Ka/Ks measure is not a reliable way to estimate the type of natural selection and there are more advanced methods for that.
